# Numerical Simulation of Effective Heat Recapture Ammonia Pyrolysis System for Hydrogen Energy

**Jian Tiong Lim [1], Eddie Yin-Kwee Ng [1,*] , Hamid Saeedipour [2,*] and Hiang Kwee Lee [3,*]**

[1] School of Mechanical and Aerospace Engineering, College of Engineering, Nanyang Technological University, Singapore 639798, Singapore; jiantion001@e.ntu.edu.sg

[2] School of Engineering, Republic Polytechnic, Singapore 738964, Singapore

[3] Division of Chemistry and Biological Chemistry, School of Physical and Mathematical Sciences, Nanyang Technological University, Singapore 637371, Singapore

[*] Correspondence: mykng@ntu.edu.sg (E.Y.-K.N.); hamid_saeedipour@rp.edu.sg (H.S.); hiangkwee@ntu.edu.sg (H.K.L.)

**Abstract:** This paper proposes a solution to address the challenges of high storage and transport costs associated with using hydrogen ($H_2$) as an energy source. It suggests utilizing ammonia ($NH_3$) as a hydrogen carrier to produce $H_2$ onsite for hydrogen gas turbines. $NH_3$ offers higher volumetric hydrogen density compared to liquid $H_2$, potentially reducing shipping costs by 40%. The process involves $NH_3$ pyrolysis, which utilizes the heat waste from exhaust gas generated by gas turbines to produce $H_2$ and nitrogen ($N_2$). Numerical simulations were conducted to design and understand the behaviour of the heat recapture $NH_3$ decomposition system. The design considerations included the concept of the number of transfer units and heat exchanger efficiency, achieving a heat recapture system efficiency of up to 91%. The simulation of $NH_3$ decomposition was performed using ANSYS, a commercial simulation software, considering wall surface reactions, turbulent flow, and chemical reaction. Parameters such as activation energy and pre-exponential factor were provided by a study utilizing a nickel wire for $NH_3$ decomposition experiments. The conversion of $NH_3$ reached up to 94% via a nickel-based catalyst within a temperature range of 823 K to 923 K which is the exhaust gas temperature range. Various factors were considered to compare the efficiency of the system, including the mass flow of $NH_3$, operating gauge pressure, mass flow of exhaust gas, among others. Result showed that pressure would not affect the conversion of $NH_3$ at temperatures above 800 K, thus a lower amount of energy is required for a compression purpose in this approach. The conversion is maintained at 94% to 97% when lower activation energy is applied via a ruthenium-based catalyst. Overall, this study showed the feasibility of utilizing convective heat transfer from exhaust gas in hydrogen production by $NH_3$ pyrolysis, and this will further enhance the development of $NH_3$ as the potential $H_2$ carrier for onsite production in hydrogen power generation.

**Keywords:** ammonia; hydrogen; power generation; pyrolysis; simulation; gas turbine

## 1. Introduction

Industrial gas turbine engines are deployed in many of today's power plants for power generation due to high thermal efficiency. To respond to the initiative of decarbonization in the power industry and beyond, new gas turbines are being developed that will be completely fuelled by $H_2$ or blend with other carbon-free fuels as an attempt to significantly reduce carbon emissions, since the combustion of $H_2$ releases almost zero carbon and sulphur oxides [1–3]. Countries worldwide are increasingly focusing on the development and implementation of hydrogen-powered gas turbine technology [4–6]. This involves the research and development of new turbine designs and the conversion of existing gas turbine power plants to run on hydrogen or hydrogen blends.

With the increasing demand of $H_2$, total $H_2$ carrier demand is expected to reach 600 million to 660 million tons by 2050, abating more than 20% of global emissions [7]. The

challenges associated with $H_2$ storage and transportation are indeed significant barriers to its widespread adoption despite its potential as a clean energy carrier. $H_2$ is an ultralight gas with 0.08375 kgm$^{-3}$ and a low boiling point of $-252.9$ °C. $H_2$ gas is widely stored under compressed gas at 35,000,000 Pa to 70,000,000 Pa or cryogenically stored at $-252.9$ °C. Hence, it would be very energy-intensive and challenging to store $H_2$. Beside this, $H_2$ pipeline has the risk of metal embrittlement that will weaken the strength of the material in the hydrogen environment, and it would result in leakage [8]. This would build up the electricity cost with the sophisticated tanks and hydrogen pipelines. The cost of shipping $H_2$ is estimated eight times higher compared with shipping $NH_3$ [9]. Therefore, an alternative $H_2$ carrier gas like $NH_3$ has been highly recommended. $NH_3$ decomposition into $H_2$ and $N_2$ can be performed via a catalytic pyrolysis process as described in Equation (1).

$$2NH_3 \leftrightarrow 3H_2 + N_2 \Delta H_{25°C} = 92 \text{ kJmol}^{-1} \tag{1}$$

$NH_3$ is the second-most produced chemical globally [10]. Therefore, infrastructure like pipelines and storage tanks are available at a large scale for implementation of onsite $H_2$ generation. The use of $NH_3$ as a $H_2$ carrier gas is hence a promising method to transport $H_2$.

However, catalytic ammonia pyrolysis is an endothermic reaction that happens at a high temperature. A typical heavy-duty gas turbine used in the power industry has a maximum of 60% efficiency of the combined cycle, which means 40% of energy input is wasted in the whole cycle. This waste energy, which accounts for the inefficiency in the turbine's operation, can be harnessed to drive the decomposition of $NH_3$ into $H_2$, which can then be directed into the combustion chamber for power generation. This integration aims to improve the overall thermal efficiency and sustainability of the gas turbine system. This energy waste can be utilized in cracking $NH_3$ to produce $H_2$ fuel and can be directed into the combustion chamber. Therefore, the gas turbine is upgraded to a self-sustaining system. As a comparison, the current $NH_3$ cracker in the market is utilizing an external heat source for $NH_3$ decomposition.

In this research study, numerical simulation has been performed on a newly designed heat recapture $NH_3$ pyrolysis system to understand the behaviour of the system and optimize the conversion of $NH_3$. In contrast to existing $NH_3$ cracking methods that rely on external heat sources, the proposed approach integrates heat recapture mechanisms within the gas turbine system itself.

## 2. Literature Review

Research on the catalytic decomposition of $NH_3$ has indeed been extensive due to its significance in various industrial processes, such as the production of fertilizers and hydrogen fuel cells. The rate-determining step in this reaction on metal catalysts like iron (Fe), cobalt (Co), or nickel (Ni) is often the desorption of $N_2$. This means that the formation of $N_2$ molecules from adsorbed species on the catalyst surface is the slowest step in the overall reaction process.

Duan et al. showed that the desorption of the nitrogen atom on the Fe surface is harder compared with that on the Co or Ni surface [11]. The nitrogen atom is present at the interstitial site and energy for $N_2$ desorption is critical here for $H_2$ generation. Hence, $N_2$ desorption is the rate-determining step in $NH_3$ decomposition [11,12]. The study explained that the ease of $NH_3$ decomposition is improved on a Ni or Co surface than on an Fe surface. Maleki et al. have modelled a flat plate microreactor to generate hydrogen by using the decomposition of $NH_3$ on a CoCeAlO mixed oxide catalyst [13]. The numerical simulations that used two different kinetic rates of reaction models showed accuracy results that were validated by experiment data [13]. The experiment revealed that the conversion of $NH_3$ hit 100% at the temperature of 550 °C. However, a Ni-based catalyst is used in this research project due to material available and cost. Beside this, Robert showed that the rate of decomposition of $NH_3$ is independent of pressure at the temperature lower than 1000 K under the application of a Ni-catalyst [14]. The study also found that the reaction

activated at the energy of $211 \pm 20$ kJmol$^{-1}$. Different support material on the Ni catalyst have different reaction rates. Ni supported by La$_2$O$_3$ have a comparable conversion of NH$_3$ with Ni supported by Al$_2$O$_3$ as the basic components in the catalyst will promote the NH$_3$ decomposition [15]. The study also indicated that the Ni-La$_2$O$_3$ catalyst allowed a higher decomposition activity with a lower weightage of Ni compared with higher weightage. Kaname et al. showed that the conversion of NH$_3$ reached 65% with a Ni-based catalyst supported by Al$_2$O$_3$ at a temperature of 500 °C [16]. Next, Chellapa et al. have introduced a suitable kinetic rate of reaction equation to be used for NH$_3$ decomposition at a temperature higher than 520 °C and a pressure above 13,332.2 Pa while using a Ni catalyst [17]. The result from their study showed that the rate of reaction is first-order dependent over the entire temperature and pressure range which is different compared to other studies [14]. Realpe et al. determined that the optimization of hydrogen production cannot rely upon only one factor, such as the catalyst, by developing a dimensionless model of the membrane reactor [18]. The approach to improve the conversion of NH$_3$ needs to be all-rounded yet not only focused on the catalyst, as the conversion of NH$_3$ is also affected by the presence of a hydrogen-selective membrane. The above studies mainly focused on the development of the catalyst in NH$_3$ decomposition. The research in the energy efficiency of NH$_3$ decomposition has received little attention, yet the process is high energy-intensive due to the endothermic reaction. In a recent study, Jaewon et al. introduced an efficiency reactor which used induction heating for green hydrogen production with an assumption of electricity generated from renewable energy [19]. However, the energy required for heating purposes would increase the usage of electricity of the plant for onsite hydrogen generation. Beside this, Yazhou et al. recently conducted a thermodynamic analysis on a heat recuperator of NH$_3$ in a humidified gas turbine that showed a net electrical efficiency of 56.7% exceeded the NH$_3$-fueled Brayton cycle by 20.6% points [20]. Hence, this suggested that a chemical recuperator of NH$_3$ in a gas turbine plant can further reduce the cost and carbon footprint in the future. Thus, a novel green hydrogen generation process that utilizes heat wastage from gas turbines is proposed in this study. The proposed process is applicable to onsite hydrogen usage for small or big gas turbine power. In this study, a numerical simulation model is used to investigate the feasibility of the process in terms of the conversion of NH$_3$ at different pressures and temperatures.

### 3. Methodology

The key technology development described here includes NH$_3$ cracking system at 100,000 Pa and at a lower temperature than 500 °C–650 °C with a suitable catalyst material. Research indicates that 100,000 Pa is the optimum condition to decompose NH$_3$ [21]. In this computational experiment, the activation energy by using Ni-based alumina catalyst will be used in the numerical design. Studies have showed that nickel alumina can obtain high conversion of NH$_3$ at around 650 °C [22]. While studies revealed that Ru-based catalyst can achieve 100% conversion at lower temperature of 350 °C [23], the rarity of ruthenium and its associated higher cost make it less suitable for large-scale projects in term of commercialization. Choosing a nickel-based catalyst also addresses the issue of nitride formation, as nickel does not tend to form nitrides during the thermal decomposition of NH$_3$ [24]. This choice contributes to the feasibility and cost-effectiveness of the proposed technology for larger-scale applications.

### 4. Heat Recapture System

This study aimed at designing a high-efficiency heat recapture system using the Number of Transfer Units (NTU) method and heat exchanger efficiency. Typically, in the context of the NTU method, the heat capacity rates of the hot and cold fluids are determined based on their mass flow rates and specific heat capacities. The NTU method is a common approach used to analyze and design heat exchangers for maximizing thermal efficiency. In addition, the NTU method is used to calculate the rate of heat transfer in heat exchangers, especially parallel flow, counter current, and crossflow exchangers. This method began

with determining the heat capacity rates of the hot and cold fluids (in $J°C^{-1}s^{-1}$) by using the Equations (2) and (3) [25]:

$$C_h = \dot{m}_h c_{ph} \tag{2}$$

$$C_c = \dot{m}_c c_{pc} \tag{3}$$

$\dot{m}_h, \dot{m}_c$ : mass flow rate of the hot and cold fluid $(kgs^{-1})$

$c_{ph}, c_{pc}$ : specific heat capacity of the hot and cold fluid $\left(Jkg^{-1}°C^{-1}\right)$

After the heat capacity rates of the hot and cold fluids have been determined, it is necessary to proceed with calculating the NTU and heat exchanger efficiency to design the heat recapture system for achieving high thermal efficiency.

Capacity ratio c is calculated by using Equation (4) after the maximum and minimum heat capacity rates have been identified from Equations (2) and (3):

$$c = \frac{C_{min}}{C_{max}} \tag{4}$$

The dimensionless group called as number of transfer units (NTU) is expressed as Equation (5):

$$NTU = \frac{UA_s}{C_{min}} \tag{5}$$

$U$ : overall heat transfer coefficienct $\left(Wm^{-2}K^{-1}\right)$

$A_s$ : heat trasnfer surface area of the heat exchanger $\left(m^2\right)$

Table 1 shows that the value of overall heat transfer coefficient can be assumed at 25 $Wm^{-2}K^{-1}$ with a practical number of tubes; this value will be cross-checked on the numerical simulation. This leads to the calculation of effectiveness of a heat exchanger which is a function of NTU and the capacity ratio, c. Different types of heat exchangers have different effectiveness calculation equations. The heat exchange type in this research is crossflow with both fluids unmixed and the effectiveness is calculated by Equation (6):

$$effectiveness = 1 - e^{\left\{\frac{NTU^{0.22}}{c}\left[e^{\left(-cNTU^{0.78}\right)} - 1\right]\right\}} \tag{6}$$

**Table 1.** Overall heat transfer coefficient and number of tubes.

| Overall Heat Transfer Coefficient | Number of Tubes |
|:---:|:---:|
| 10 | 49 |
| 25 | 19 |
| 50 | 9 |
| 75 | 6 |
| 100 | 4 |

The effectiveness values and NTU values are obtained from ([25], pp. 705–706) to identify the effectiveness and the length of the tubes in the heat exchanger [25]. The thermal efficiency of the design is calculated by Equation (7):

$$thermal\ efficiency = \frac{T_{NH_{3ave}} - T_{NH_{3in}}}{T_{hotgas_{in}} - T_{NH_{3in}}} \times 100\% \tag{7}$$

$T_{NH_{3ave}}$ : $NH_3$ volumetric average temperature(K)
$T_{NH_{3in}}$ : $NH_3$ inlet temperature(K)
$T_{hot\ gas_{in}}$ : Hot gas inlet temperature (K)

Design of the heat recapture system as shown in Figure 1 has been developed by using NTU and effectiveness method. It had a dimension of 320 mm long with 120 mm diameter. It had 19 tubes with 12 mm external diameter and thickness of 1 mm. $NH_3$ inlet and $NH_3$ outlet are set at 5 mm diameter.

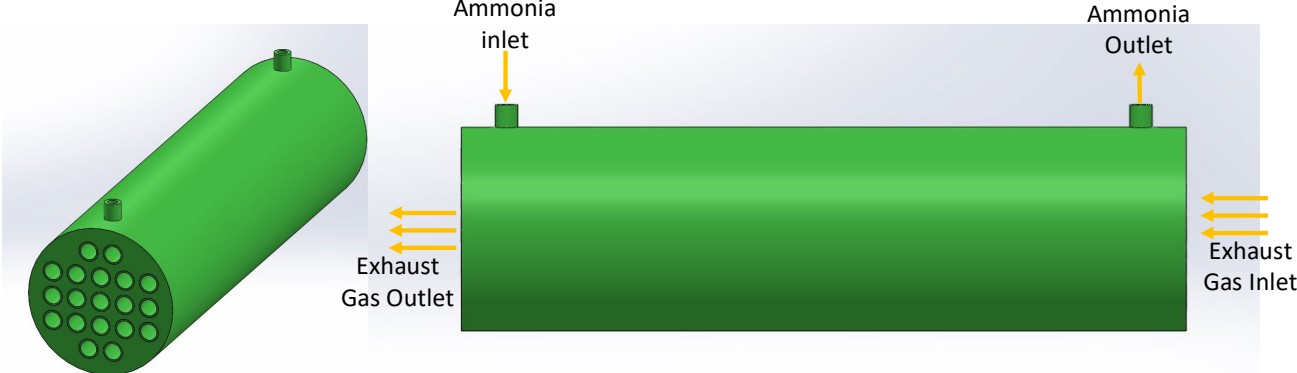

**Figure 1.** Design of heat recapture system.

ANSYS Fluent 2019 R1 software from Ansys, Canonsburg, US has been used for numerical simulation to validate theoretical calculations and perform a mesh independence analysis to ensure accurate results with low computational resources [26]. The computational analysis of the heat recapture system has been performed by using ANSYS Fluent to verify the theory calculation with the boundary conditions in Table 2. A grid-independent analysis is performed on the 3D geometry that could provide an accurate result with low computational time and cost. A table of mesh element size vs. other parameters is shown in Table 3. The mesh selected size is 8 mm with mesh node of 655,819 and mesh element of 1,206,553. Realizable $k - \varepsilon$ model, standard $k - \omega$ model and shear stress transport (SST) $k - \omega$ model are applied and their performance on this flow simulation is compared. Realizable $k - \varepsilon$ model is selected to determine the overall thermal efficiency of the entire system as $k - \omega$ model consumed lower computational cost compared with $k - \varepsilon$ model [27]. All the governing equations were solved using the commercial simulation software ANSYS Fluent 2019 R1.

**Table 2.** Values of each parameter.

| Parameter | Value/Description |
|---|---|
| Exhaust Gas Inlet Temperature | 650 °C |
| Exhaust Gas Inlet Velocity | 12 m/s |
| $NH_3$ Inlet Temperature | 30 °C |
| $NH_3$ Inlet Velocity | 4 m/s |
| Material | Stainless Steel (SS316) |

**Table 3.** Grid-independence test result for thermal simulation.

| Mesh Element Size (mm) | Number of Mesh Node | Number of Mesh Elements | $NH_3$ Average Temperature (K) | $NH_3$ Average Turbulent Kinetic Energy (J/kg) | $NH_3$ Effective Prandtl Number |
|---|---|---|---|---|---|
| 20 | 655,461 | 1,204,453 | 865.1 | 0.0161 | 0.8908 |
| 10 | 655,399 | 1,203,915 | 865.7 | 0.0161 | 0.8909 |
| **8** | **655,819** | **1,206,553** | **867.5** | **0.0161** | **0.8909** |
| 5 | 654,496 | 1,202,314 | 865.4 | 0.0161 | 0.8908 |

### 5. NH$_3$ Decomposition

A kinetic model that could express the rate of NH$_3$ decomposition reaction can be defined as Equation (8):

$$R_{NH_3} = k_{NH_3} \left[ \left( \frac{C_{NH_3}^2}{C_{H2}^3} \right)^\beta \right] \tag{8}$$

where $C_{NH_3}$ is the concentration of NH$_3$, $C_{H_2}$ is the concentration of hydrogen, $\beta$ is a fit parameter, $k_{NH_3}$ is the rate constant of NH$_3$ decomposition [13]. This equation is based on the Temkin–Pyzhev model. Hence, beside the rate constant, the value of $\beta$ has great impact on the rate of decomposition. Studies showed that the value of $\beta$ approached unity with lower $\Delta H$ and approached zero at higher $\Delta H$. $\beta$ is set at 0.5 in this study as learned from [12].

A study conducted by Robert et al. on decomposition of NH$_3$ on nickel wire has been used as a reference for validation purposes in this study [14]. The study suggested that the rate constant for NH$_3$ pyrolysis under 1000 K is approximated by Equation (9):

$$k_{NH_3} = k^0 e^{-\frac{E}{RT}} \tag{9}$$

$k^0 = 1 \times 10^{13} \text{ s}^{-1}$
$E = 211 \text{ kJmol}^{-1}$
$R = 8.314 \text{ Jmol}^{-1}\text{K}^{-1}$
$T = \text{NH}_3 \text{ temperature (K)}$

The above values will be set in the ANSYS Chemkin Pro 2019 R1 software. For comparison purposes, 105 kJmol$^{-1}$ and 89.11 kJmol$^{-1}$ are set as the activation energy, respectively, as study suggested that the activation energy for NH$_3$ decomposition has a range of 211 kJmol$^{-1}$ to 25 kJmol$^{-1}$ [14,28]. The rate constant k used in ANSYS Chemkin Pro 2019 R1 software is described as Equation (10):

$$k = AT^B e^{-\frac{E}{RT}} \tag{10}$$

where B is temperature exponent (dimensionless), A is pre-exponential factor, and E is activation energy $\left( \text{kJmol}^{-1} \right)$. The rate of reaction R in ANSYS Chemkin Pro 2019 R1 software is described as Equation (11):

$$R = k \left( \prod_{i=1}^{N_g} C_i^{n_i} \right) \tag{11}$$

where $N_g$ is the total number of gas species, $C_i$ is the concentration of each species ith as reactant in the reaction $\left( \text{moldm}^{-3} \right)$, and $n_i$ is the rate exponent for the ith each species. The conversion of NH$_3$ obtained from this simulation will be calculated by using the following equation:

$$\text{NH}_3 \text{ conversion\%} = \frac{2\gamma_{N_2}}{2\gamma_{N_2} + \gamma_{NH_3}} \times 100\% \tag{12}$$

where $\gamma_i$ is the mole fraction of the species i.

Other equations involved in mass transfer effect can be referred in [29]. A numerical simulation has been conducted to verify the study conducted by Robert et al. so that it verified that this simulation setup in this study is accurate [14]. This study investigated the reaction rate of NH$_3$ decomposition at 4 different operating gauge (or barometric) pressures which are 5.3 Pa, 13.3 Pa, 53 Pa, and 133 Pa. The simulation setup for validation is a 2D simulation represented in Figure 2. Grid-independent test has been conducted shown in Figure 3; the average rate of reaction reached the lowest after the mesh size reduced to

5 mm and a mesh element size of 5 mm with 1728 mesh nodes, and 1643 mesh elements have been selected as tetrahedral meshing method is used.

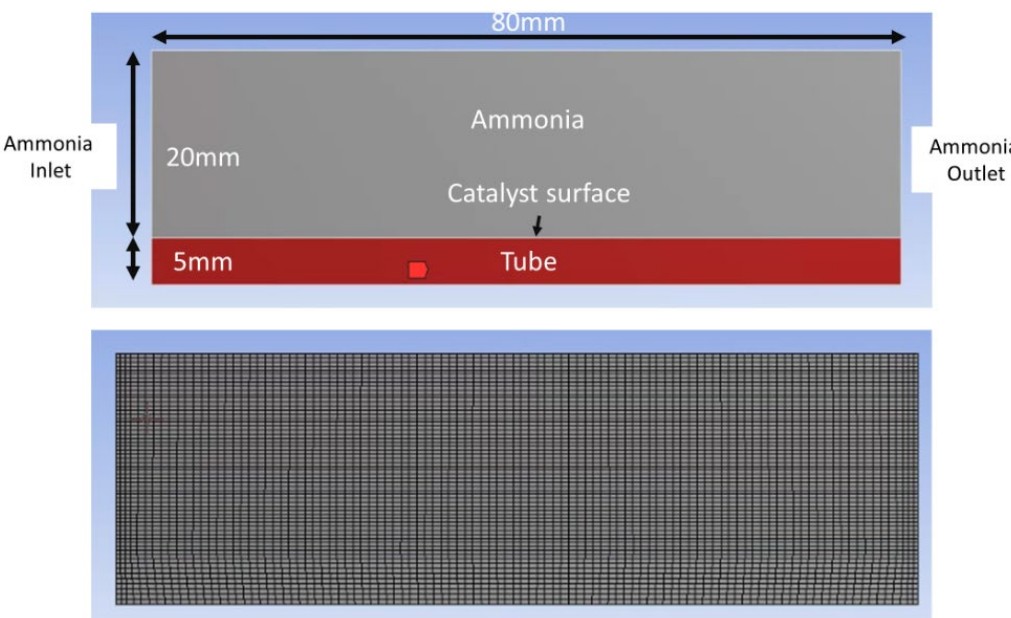

**Figure 2.** 2D Simulation setup for study validation.

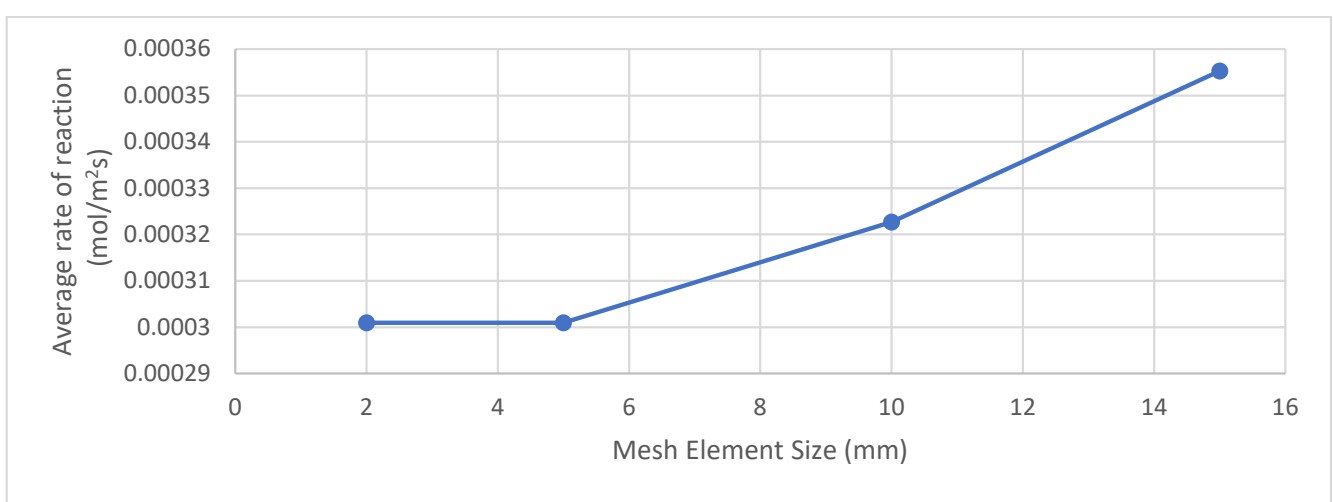

**Figure 3.** Grid-independent test: mesh element size vs. average rate of reaction graph.

The results suggest that the rate of reaction exhibits different dependencies on temperature and pressure within distinct temperature ranges at $NH_3$ inlet velocity of 5 m/s in Figure 4. Below 723 K, the reaction rate demonstrates a first-order dependence on temperature, while exhibiting zero-order dependence on pressure. However, at temperatures exceeding 723 K, the reaction rate becomes first-order dependent on pressure.

Moreover, the results indicate that at temperatures above 923 K, the reaction rate increases with higher pressure. Specifically, as shown in Figure 4, at 133 Pa, the rate of reaction is approximately 1.3 times higher compared to the rate at 53 Pa. Similarly, at 13.3 Pa, the rate is approximately 1.4 times higher than at 5.3 Pa. These findings are consistent with those reported in [14], reinforcing the validity of the numerical simulation in this study. It shows that numerical simulation can be used to investigate the chemical reaction in $NH_3$ decomposition.

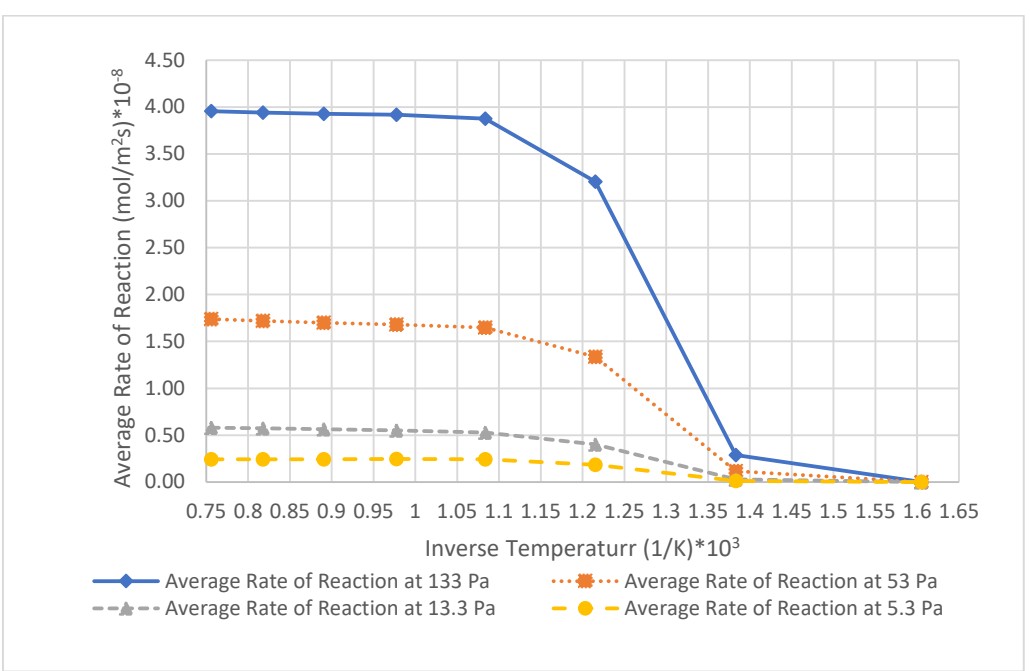

**Figure 4.** Inverse temperature vs. average rate of reaction graph at different pressure.

Moreover, the results indicate that at temperatures above 923 K, the reaction rate increases with higher pressure. Specifically, as shown in Figure 4, at 133 Pa, the rate of reaction is approximately 1.3 times higher compared to the rate at 53 Pa. Similarly, at 13.3 Pa, the rate is approximately 1.4 times higher than at 5.3 Pa. These findings are consistent with those reported in [14], reinforcing the validity of the numerical simulation in this study. It shows that numerical simulation can be used to investigate the chemical reaction in $NH_3$ decomposition.

In this study, numerical simulation has been applied to investigate the performance of the heat recapture $NH_3$ pyrolysis system under different operating conditions via convective heat transfer, while making necessary simplifications to manage computational resources effectively.

The grid-independence test has been applied in these numerical simulations where the mesh size was varied to ensure that the results of the simulation do not significantly change with different mesh sizes. It is important to establish grid independence to ensure the accuracy of the numerical simulation results.

It is assumed that the catalyst temperature remains constant across all surfaces. While this simplifies the simulation, it is important to note that the catalyst temperature may vary across different parts of the system. To reduce computational cost and time, it was decided not to simulate the flow of hot gas in the system. This simplification is common in simulations where the focus is on specific aspects of the system, such as the reaction kinetics or heat transfer within the catalyst bed. The performance of the system under different catalyst temperature conditions has been compared, ranging from 623 K to 923 K to study how changes in catalyst temperature affect the behaviour of the system and its efficiency in $NH_3$ pyrolysis.

In the numerical simulation of the newly designed heat recapture $NH_3$ pyrolysis system, grid-independent test has been conducted on this design shown in Figure 5. The 0.01 m mesh element size is chosen with a total of 842,165 elements and 185,873 mesh nodes as shown in Figure 6.

Planes at three sections, inlet, mid-length, and outlet shown in Figure 7, are used to simulate $NH_3$ gas temperature, $NH_3$ gas velocity, and mass fraction of each species. ZY plane in Figure 8 is used to simulate $NH_3$ temperature contour and mass fraction of $H_2$ at different $NH_3$ inlet velocity.

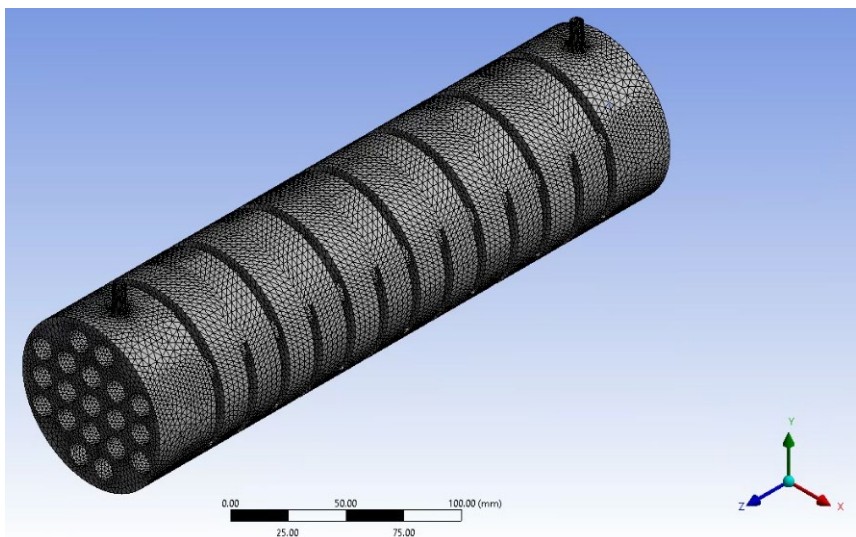

**Figure 5.** 3D simulation of $NH_3$ pyrolysis system.

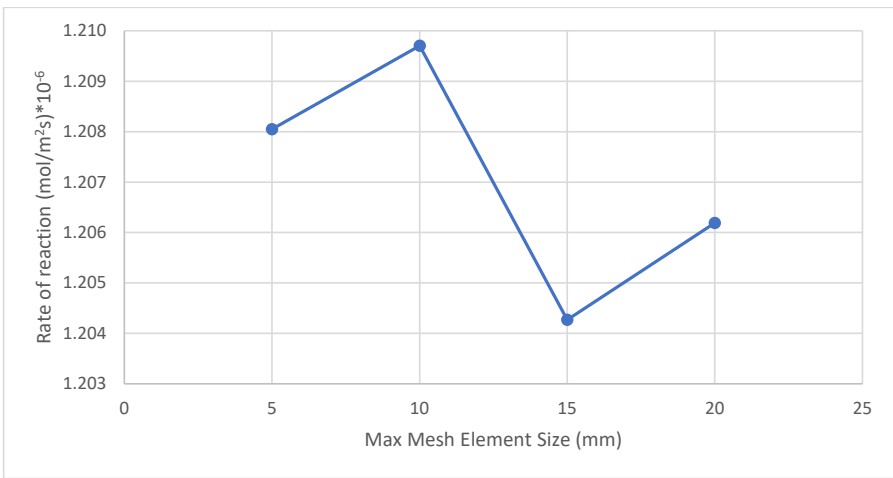

**Figure 6.** Grid-independent test result for 3D $NH_3$ decomposition simulation.

Planes at three sections, inlet, mid-length, and outlet shown in Figure 7, are used to simulate $NH_3$ gas temperature, $NH_3$ gas velocity, and mass fraction of each species. ZY plane in Figure 8 is used to simulate $NH_3$ temperature contour and mass fraction of $H_2$ at different $NH_3$ inlet velocity.

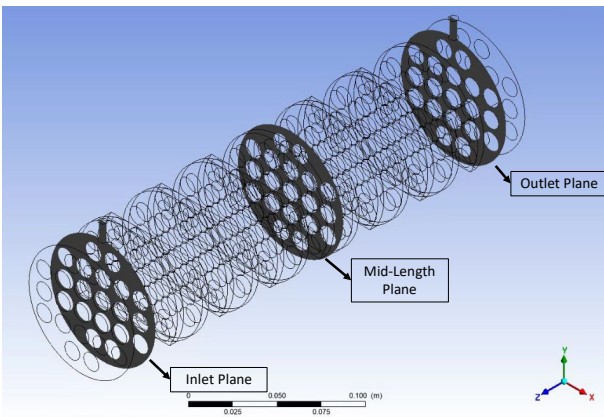

**Figure 7.** 3 Different section planes of the $NH_3$ pyrolysis system.

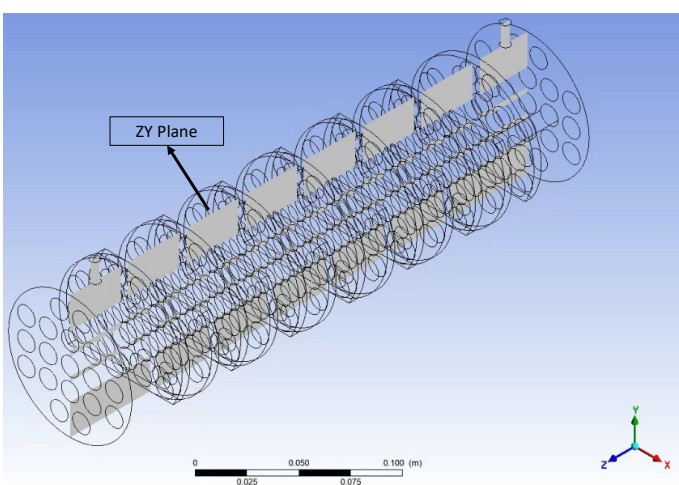

**Figure 8.** ZY plane of the NH$_3$ pyrolysis system.

## 6. Results and Discussion

### 6.1. Heat Recapture System

The realizable k − ε model, SST k − ω model, and standard k − ω model have been used in the thermal simulation for the heat recapture NH$_3$ pyrolysis system. Three of the models gave closed results as shown in Figure 9. The NH$_3$ average temperature that is taken from the volumetric average of the NH$_3$ temperature in the system is used as the comparison parameter. So, for a more efficient and conservative result, a realizable k − ε model has been selected as it has lower computational cost and time compared with the other two models.

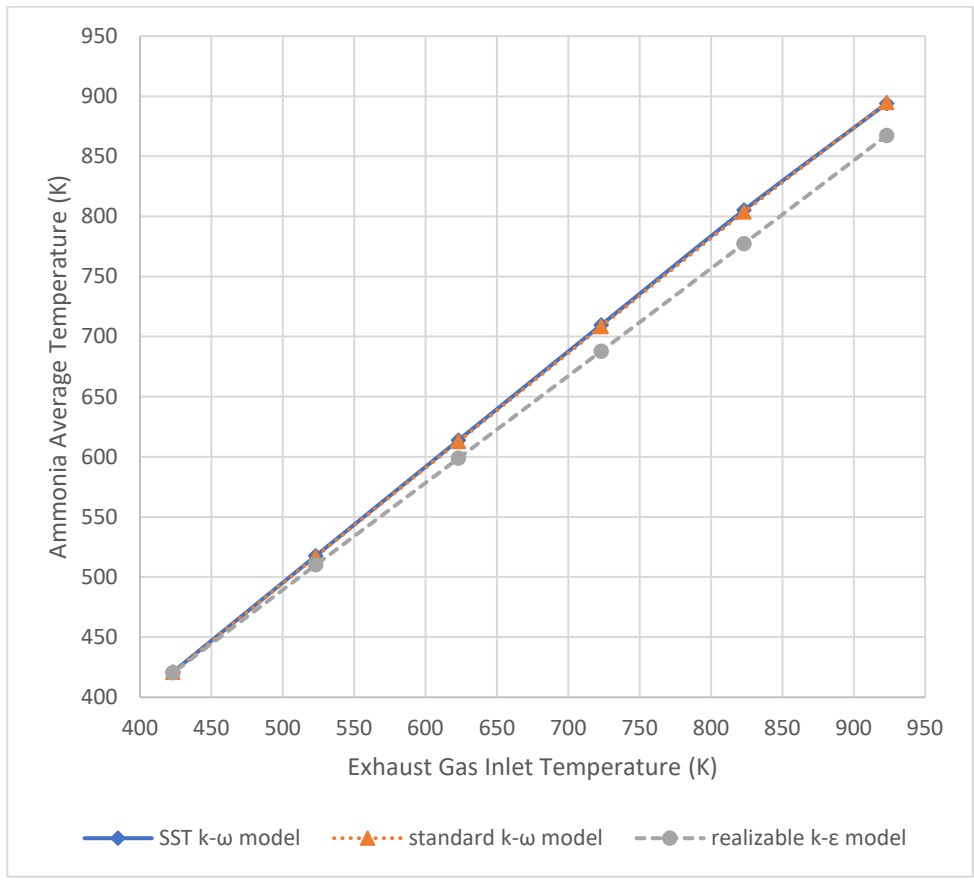

**Figure 9.** Exhaust gas inlet temperature vs. average NH$_3$ temperature graph.

Based on the designed system, a numerical simulation showed that the higher the exhaust gas velocity, the better the efficiency. Our calculation that was based on the NTU method revealed that the system could achieve 83.87% but the simulation indicated near to 91% at 4 ms$^{-1}$ of NH$_3$ velocity plotted in Figure 10. This is due to the assumption of laminar flow in the NTU method; likewise, the realizable k $-$ ε model in the simulation took turbulent flow into consideration, hence efficiency could be higher due to turbulent enhancement on heat transfer and lesser pressure loss.

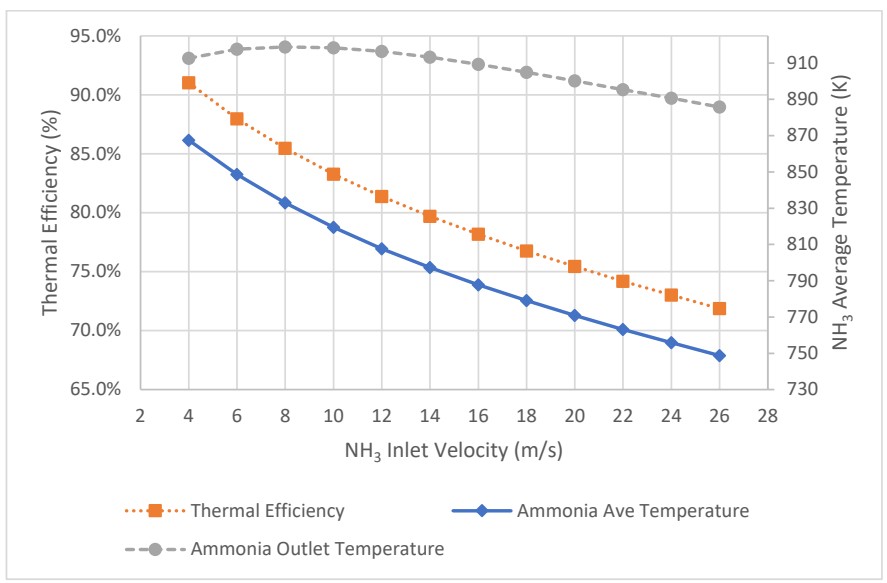

**Figure 10.** NH$_3$ Inlet velocity vs. thermal efficiency and NH$_3$ average temperature.

High exhaust gas inlet velocity resulted in high thermal efficiency in the system as presented in Figure 11 since high mass flow of high temperature will enhance the convective heat transfer from exhaust gas to NH$_3$ gas. This could further reduce the temperature of the exhaust gas if there is a surge in the gas turbine after the heat recapture system and release to the surrounding area. Hence, it eliminates the need of a cooling system downstream from the gas turbine before the release of the exhaust gas.

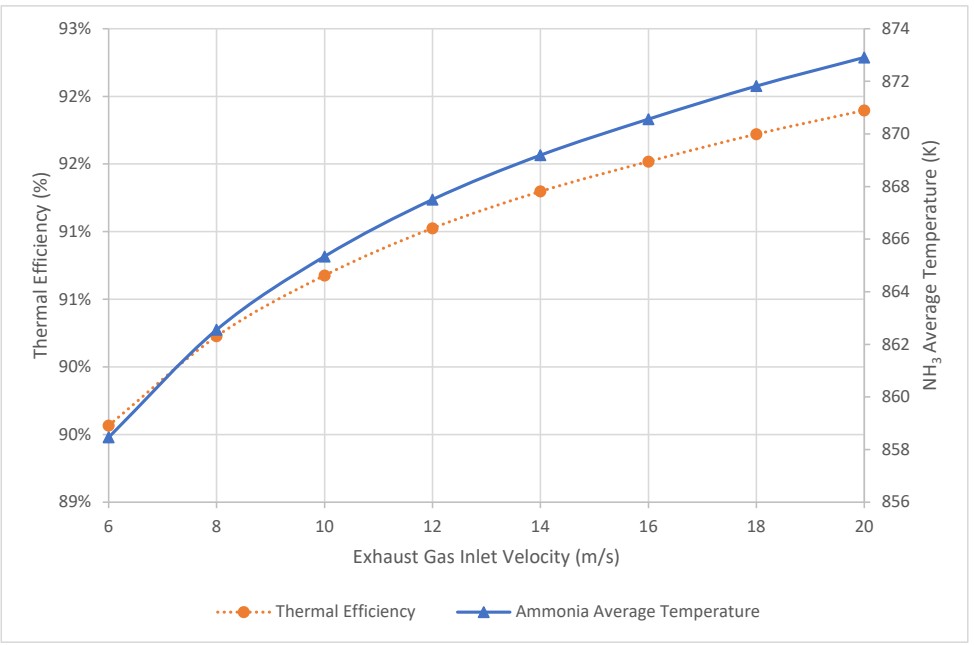

**Figure 11.** Exhaust gas inlet velocity vs. thermal efficiency and NH$_3$ average temperature.

Turbulent kinetic energy is high at the inlet section presented in Figure 12 as swirl flows occur at the inlet section when $NH_3$ gas is pumping in. It is observed that turbulent intensity is higher at the area where swirl flows occurred as shown in Figure 13, and swirl flows are distributed more closely at the gap between tubes. Hence, the mixture of $NH_3$ flow at the swirl flows could increase the thermal efficiency of $NH_3$ by enhancing the convective heat transfer [30] as swirl flows will increase the efficiency of the heat transfer of $NH_3$ and the catalyst tubes. Temperature contour in Figure 14 shows that more than half of the $NH_3$ volume at temperature of 904 K and this resulted in high total enthalpy gained by $NH_3$ plotted in Figure 15. $NH_3$ gained a maximum of $1.66 \times 10^6$ $Jkg^{-1}$ total enthalpy. By assuming the activation energy is 211 $kJmol^{-1}$ for $NH_3$ decomposition, this enthalpy could convert 0.1 mol of $NH_3$ to 0.15 mol of hydrogen and 0.05 mol of nitrogen or 0.18 mass fraction of hydrogen and 0.92 mass fraction of nitrogen for complete conversion of $NH_3$. This result will be validated by the simulation result for $NH_3$ decomposition in the next section.

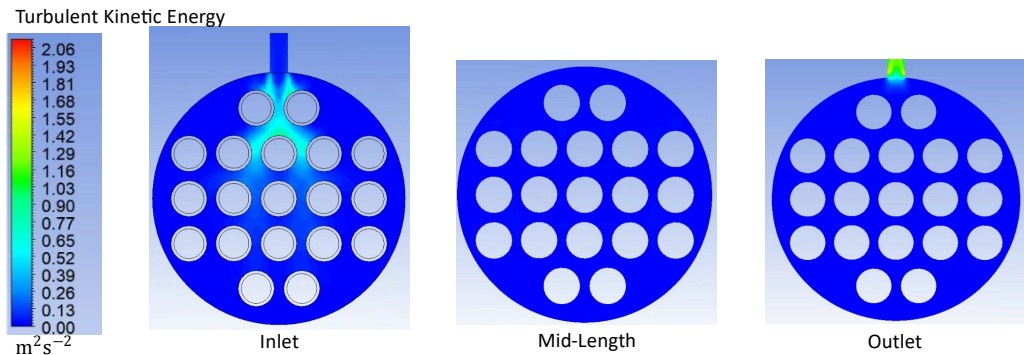

**Figure 12.** Turbulent kinetic energy contour of $NH_3$ gas at 3 different sections.

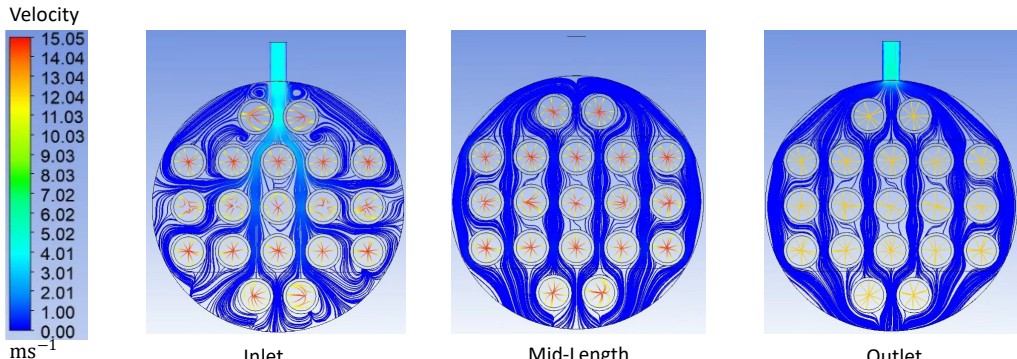

**Figure 13.** Velocity contour of $NH_3$ gas at 3 different sections.

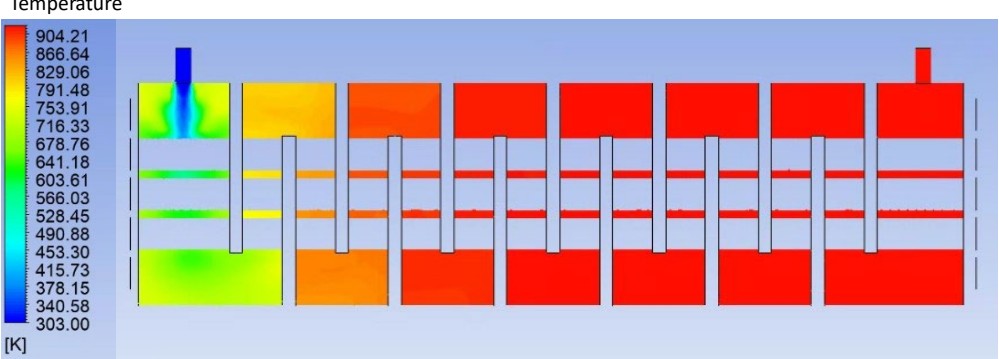

**Figure 14.** Temperature contour of $NH_3$ gas of ZY plane.

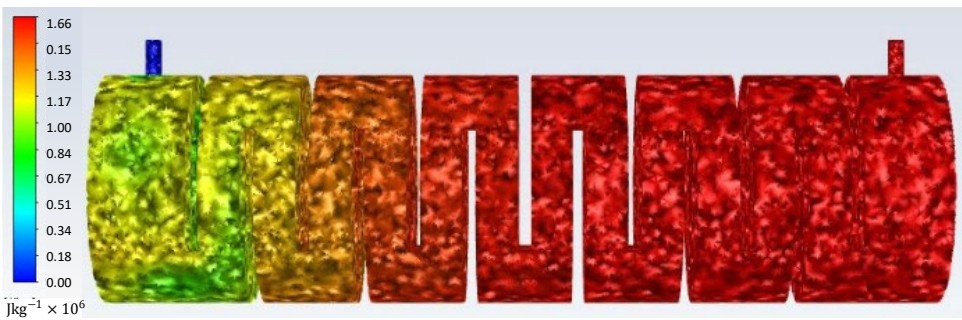

**Figure 15.** Total enthalpy contour of $NH_3$ gas in the system.

### 6.2. Heat Recapture $NH_3$ Pyrolysis System

Numerical simulation suggested that the rate of reaction of $NH_3$ pyrolysis is at first-order temperature dependent at temperature lower than 723 K and it changed from zero-order to first-order pressure dependent at temperature above 723 K as shown in Figures 16–18. The finding from this study is different from the observation made by [14] as the pressure dependent changed from zero to first order at temperature above 1000 K. Therefore, the heat capture $NH_3$ pyrolysis system has lowered the temperature for the pressure dependent and achieved a higher rate of reaction at a lower temperature. Our designed system has the highest rate of reaction at 823.15 K with a pressure of 300,000 Pa and $NH_3$ inlet velocity of 8 ms$^{-1}$. This could simply be explained by the effect of mass flow of the $NH_3$ on the rate of $NH_3$ decomposition. The rate of reaction seemed to reach the peak at 300,000 Pa between temperatures of 723 K and 923 K. This range of temperature is considered as the working temperature for $NH_3$ pyrolysis in this system as exhaust gas from a typical power plant gas turbine is released at temperatures between 823.15 K and 895.15 K.

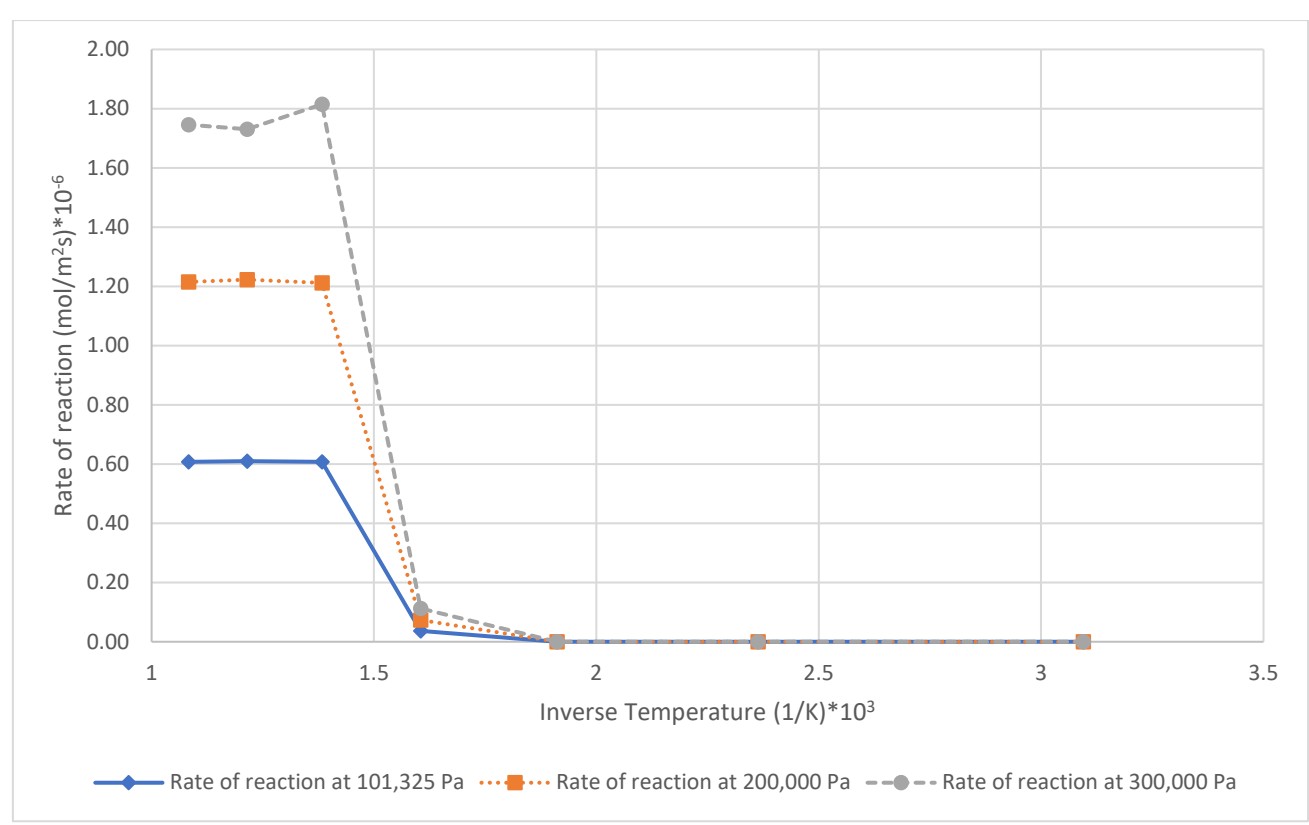

**Figure 16.** Inverse temperature vs. rate of reaction at $NH_3$ inlet velocity of 2 ms$^{-1}$.

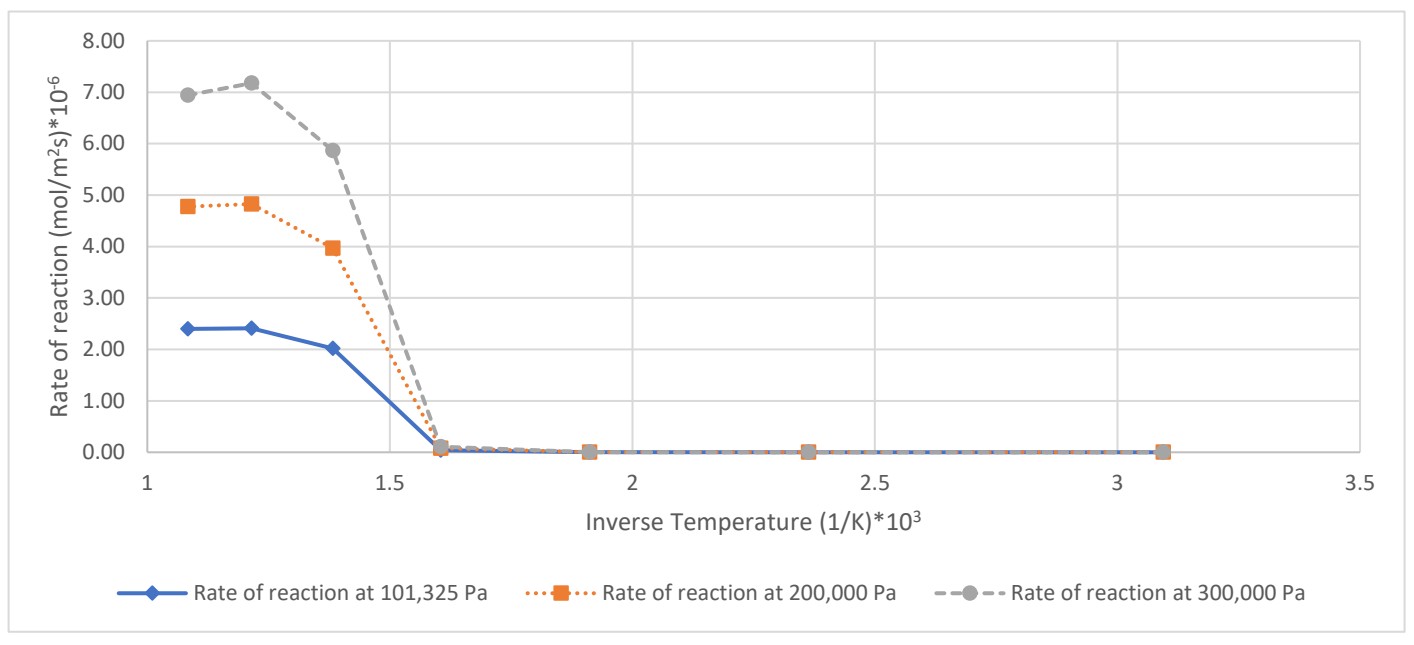

**Figure 17.** Inverse temperature vs. rate of reaction at NH$_3$ inlet velocity of 4 ms$^{-1}$.

**Figure 18.** Inverse temperature vs. rate of reaction at NH$_3$ inlet velocity of 8 ms$^{-1}$.

Figures 16–18 outline the behaviour of the conversion of NH$_3$ under varying conditions of temperature and pressure, as well as the performance of a designed system within specific temperature ranges. The trends in these graphs show the influence of pressure

and temperature on the conversion of $NH_3$, as well as the successful performance of a designed system within a specific temperature range. The rate of reaction in this study is 1st order dependent of temperature but zero order dependent of pressure below 623.15K. Dependent of pressure changed to 1st order above temperature of 623.15K. The conversion of $NH_3$ decreased at a constant temperature with increasing pressure at temperature below 823 K and this is supported by study [31]. There is no conversion of $NH_3$ at temperature below 623 K where a reaction failed to take place as shown in Figure 19. Conversion of $NH_3$ remained the same at temperature above 800 K at different pressures. Thus, this could reduce the energy required from compression purposes. The designed system achieved near to 94% conversion of $NH_3$ at the temperature ranges from 823 K to 923 K which is the working temperature of the system. One of the possible reasons that the system failed to reach 99% conversion could be the assumption made at the $NH_3$ outlet condition. The $NH_3$ outlet is assumed to allow the backflow of non-decomposed $NH_3$. Beside this, it is important to note that the calculation of the conversion of $NH_3$ is taken with the average mole fraction of each species.

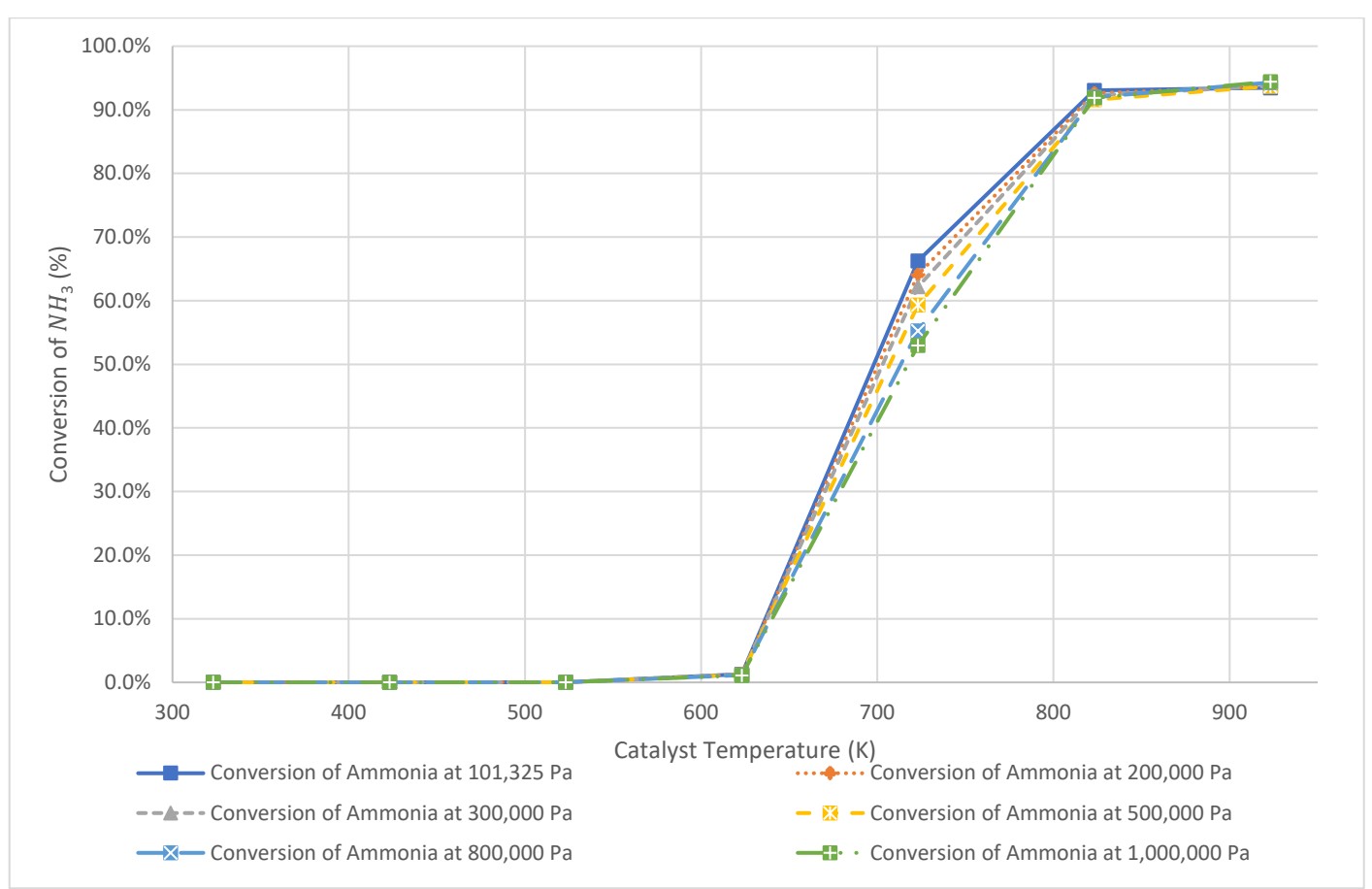

**Figure 19.** Catalyst temperature vs. conversion of $NH_3$.

At $NH_3$ inlet velocity of 4 ms$^{-1}$, $NH_3$ begins to be decomposed at the inlet section as seen in Figure 20. $NH_3$ gas gained enough energy near the catalyst tubes and hence, $NH_3$ started to decompose into hydrogen and nitrogen around the tubes' surfaces as catalyst $NH_3$ pyrolysis is a wall surface reaction [32]. The mass fraction of $H_2$ reaches up to 0.18 at the mid-plane and outlet plane shown in Figures 21 and 22 which are consistent with the mole concept calculation based on the total enthalpy gained by the $NH_3$ gas and the activation energy of 211 kJmol$^{-1}$. The theoretical calculation involved is explained in Appendix A to verify the validity of the result obtained from numerical simulation.

Mass Fraction of Each Species at Inlet Plane

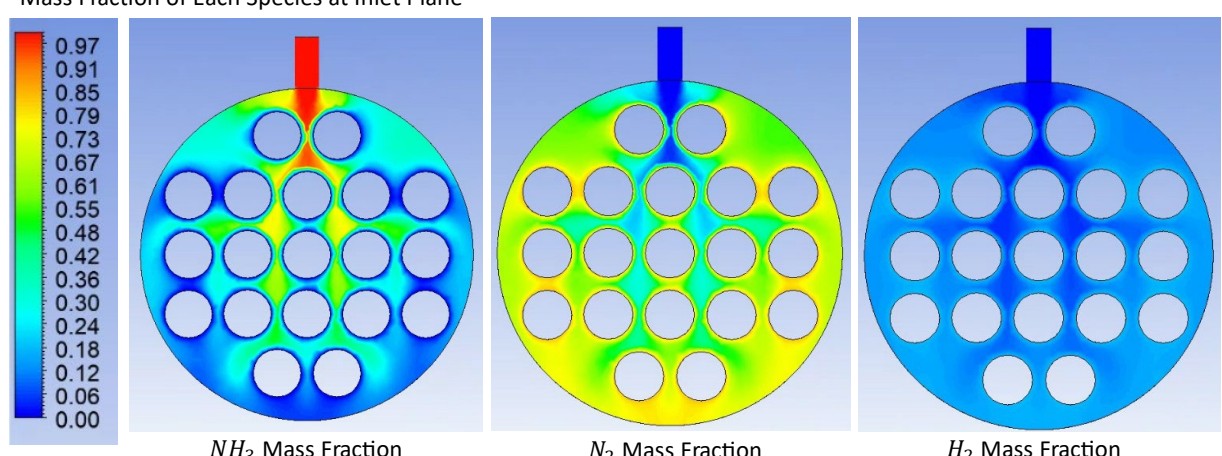

**Figure 20.** Mass fraction of each species at inlet plane.

Mass Fraction of Each Species at Mid Plane

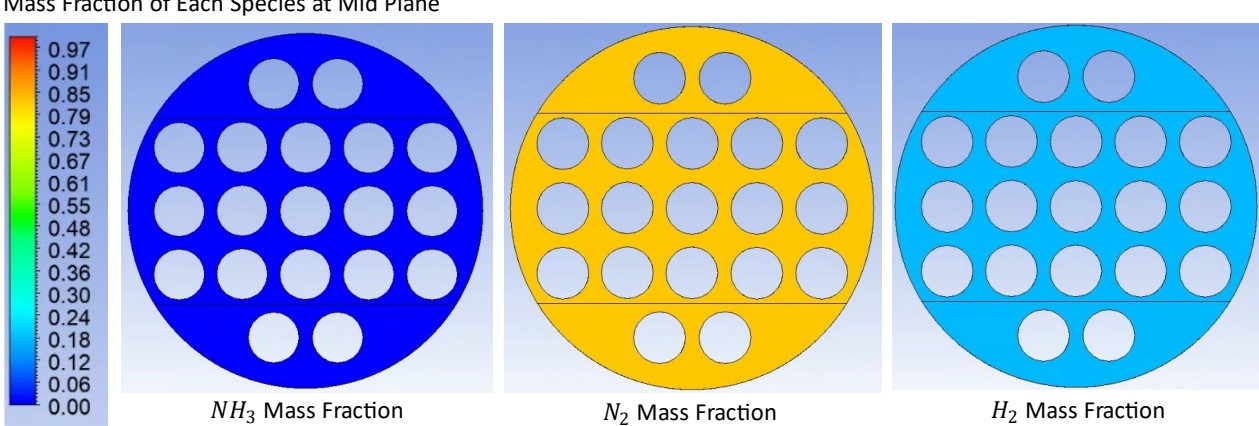

**Figure 21.** Mass fraction of each species at mid-length.

Mass Fraction of Each Species at Outlet Plane

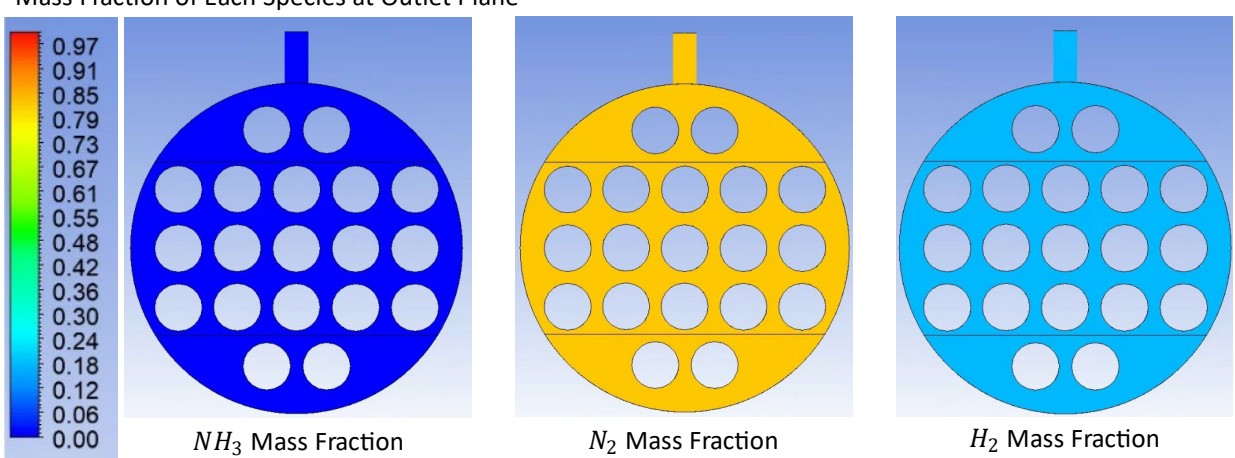

**Figure 22.** Mass fraction of each species at outlet plane.

NH$_3$ mass flow is required for high hydrogen generation in the system. However, the higher the NH$_3$ inlet velocity, the lower the conversion of NH$_3$ as shown in Figures 23 and 24. The contour area for 0.18 mass fraction of H$_2$ is reducing with the increase of NH$_3$ inlet velocity. NH$_3$ gas escaped from the system before it gained enough energy for

decomposition to take place. The maximum velocity of the $NH_3$ inlet as it provided the maximum thermal efficiency is 4 ms$^{-1}$ in this study and the highest conversion of $NH_3$ when the exhaust gas inlet velocity is 12 ms$^{-1}$. The possible way to increase the effective mass flow of $NH_3$ is to increase the exhaust gas inlet velocity as this will certainly improve the convective heat transfer from Figure 11. The exhaust gas inlet velocity is reduced by a diffuser downstream the gas turbine to slow down the exhaust gas velocity. The design of the diffuser would thus be one of the critical parts in system integration of heat recapture $NH_3$ pyrolysis with a gas turbine.

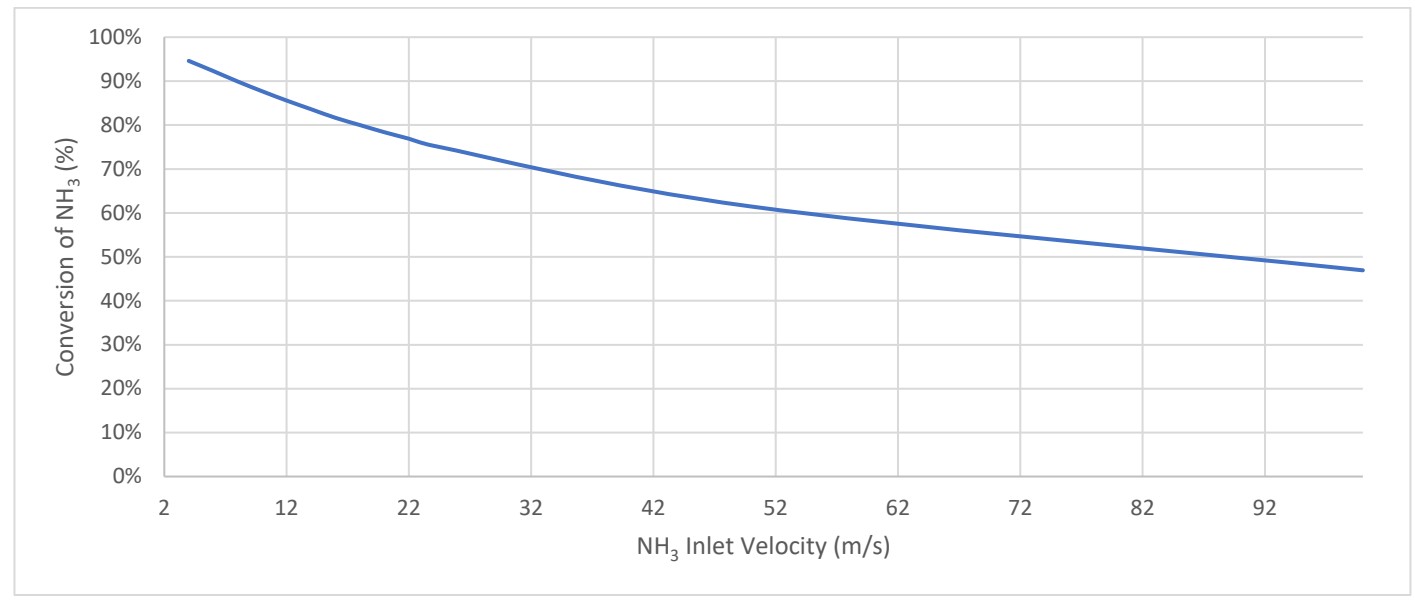

**Figure 23.** $NH_3$ Inlet velocity vs. conversion of $NH_3$.

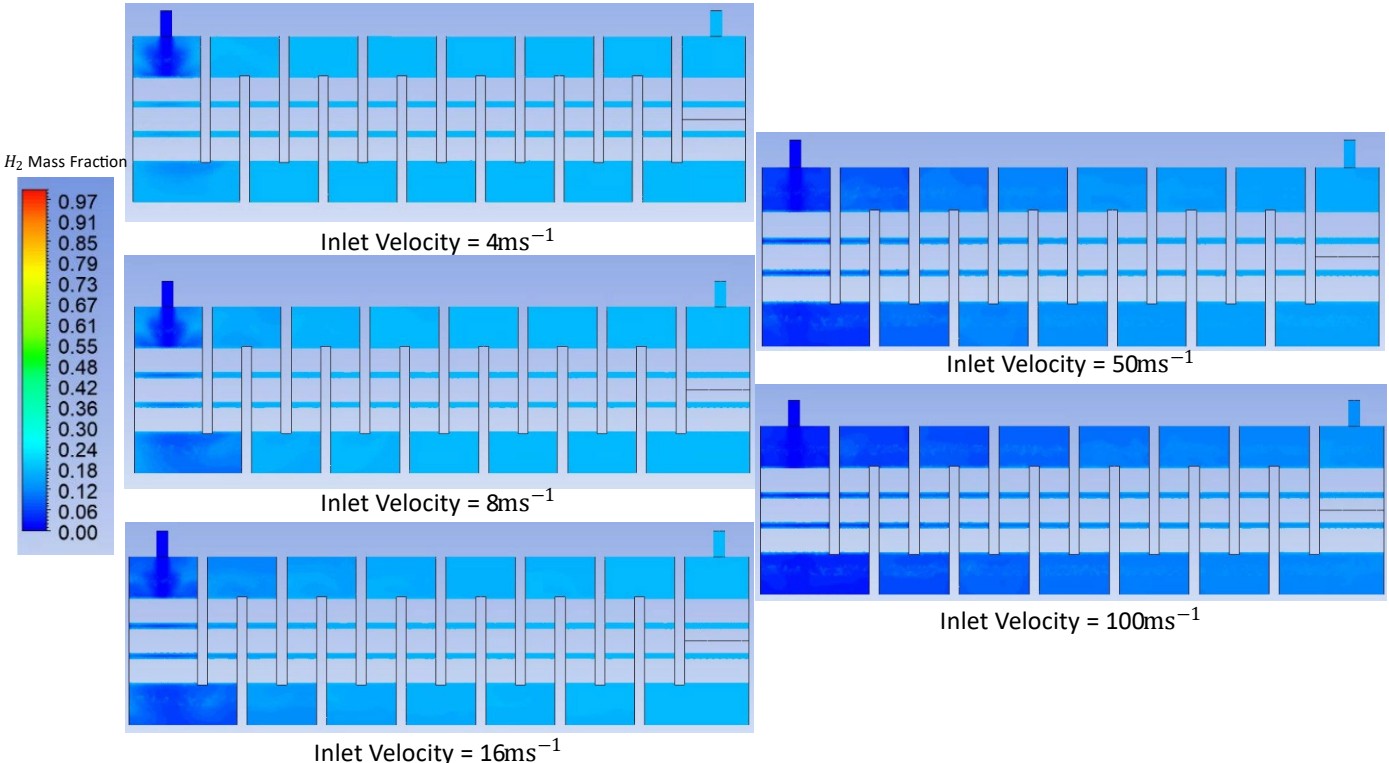

**Figure 24.** Contour of mass fraction of $H_2$ at different $NH_3$ inlet velocity.

Three different activation energies are used in the numerical simulation and the conversion of $NH_3$ on these energies is plotted in Figure 25. The graph suggests that the lower the activation energy, the higher the conversion of $NH_3$. A lower activation energy also allowed $NH_3$ to be decomposed at a lower temperature. Hence, the coatings of the catalyst and the catalyst material are important factors here. The coating of the catalyst can affect the surface binding and heat transfer of the tubes and the catalyst. Catalysts like a Ni-based catalyst will give a higher activation energy. Catalysts like a ruthenium-based catalyst and a platinum-based catalyst will give a lower activation energy [33].

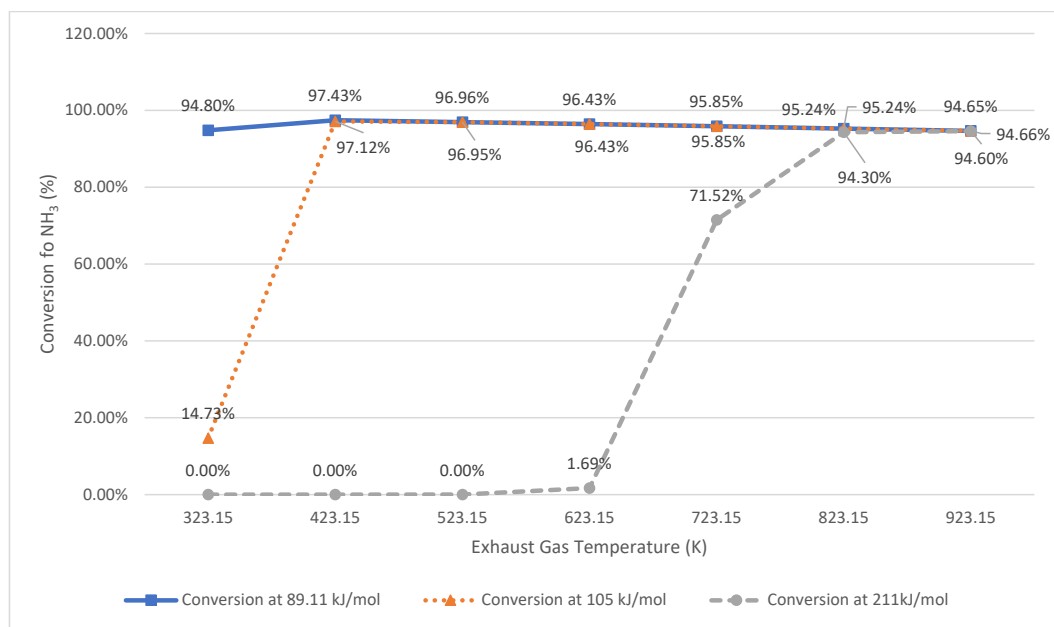

**Figure 25.** Exhaust gas temperature vs. conversion of $NH_3$ at different activation energies.

Last but not least, the temperature exponent, B, is one of the important dimensionless parameters in the rate of reaction equation. However, no study has been conducted to determine the suitable value of the temperature exponent for the $NH_3$ decomposition rate of reaction but the temperature exponent could certainly affect the rate of reaction shown in Figure 26. The shift is from a near 0 rate of reaction to the peak of the rate of reaction at $1.2 \times 10^{-6}$ $m^{-2}s^{-1}$mol at the temperature of 623 K. Hence, the temperature exponent could be one of the parameters to be tuned in an $NH_3$ pyrolysis system design either via electrolysis or heat recapture.

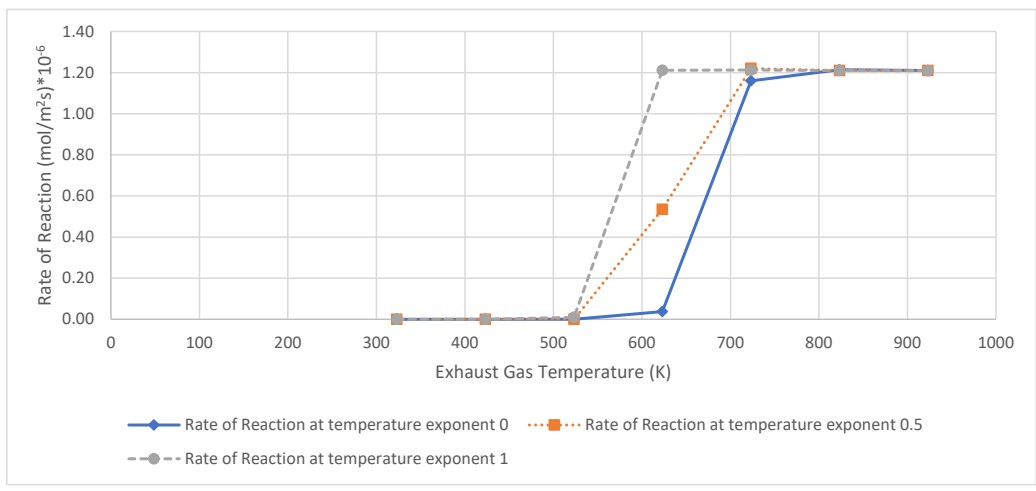

**Figure 26.** Exhaust gas temperature vs. rate of reaction at different temperature exponents.

### 7. Conclusions

NH$_3$ pyrolysis presents a promising avenue for sustainable hydrogen production, but its efficiency can be significantly enhanced through the integration of heat recapture systems. In this study, we present a numerical simulation of a heat recapture NH$_3$ pyrolysis system, aiming to evaluate its performance and potential for industrial application. The simulation framework encompasses thermodynamic modelling of the pyrolysis reactor, heat exchangers, and associated components. Utilizing advanced computational fluid dynamics techniques, we analyzed the heat transfer mechanisms and optimized system parameters to maximize energy efficiency. Our results demonstrate the effectiveness of heat recapture strategies in improving overall process efficiency and reducing environmental impact. This research contributes to the advancement of sustainable hydrogen production technologies and offers insights for the design and operation of NH$_3$ pyrolysis systems in real-world applications.

The shift towards hydrogen-powered gas turbines is crucial for decarbonizing the power industry and mitigating climate change. Despite its environmental benefits, challenges in storage and transportation hinder widespread adoption. This paper explores using NH$_3$ as a hydrogen carrier, leveraging its higher energy density and existing infrastructure. NH$_3$ decomposition, utilizing waste heat from gas turbines, offers a promising solution. The conversion process involves technical and logistical challenges but promises significant emission reductions. The global demand for H$_2$ carriers is rising, emphasizing the need for alternative solutions. This paper proposes integrating NH$_3$ decomposition within gas turbine systems to enhance efficiency and sustainability.

An extensive literature survey has been conducted on catalytic NH$_3$ decomposition, highlighting its importance in various industrial processes. Studies demonstrate the significance of catalysts like nickel in achieving high conversion rates. Various factors, including catalyst composition and temperature, influence NH$_3$ decomposition kinetics. Numerical simulations and experimental validations provide insights into reaction mechanisms and system performance. Previous research emphasizes the potential of NH$_3$ as a hydrogen carrier and its integration within gas turbine systems to improve efficiency.

The transition to H$_2$ as an energy source faces challenges related to high storage and transportation costs. This paper proposes a solution using NH$_3$ as a hydrogen carrier, offering higher volumetric hydrogen density and potential shipping cost reduction by 40%. This study focuses on designing a high-efficiency NH$_3$ cracking system using a nickel-based catalyst. Numerical simulations and theoretical calculations are performed to optimize system design and understand reaction kinetics. The heat recapture system is designed using the NTU method and heat exchanger efficiency considerations. ANSYS simulations validate theoretical calculations and analyze system behaviour under different operating conditions. The NH$_3$ decomposition kinetics are modelled based on experimental data, considering factors such as temperature and pressure dependencies. NH$_3$ pyrolysis, utilizing waste heat from gas turbines, can produce H$_2$ onsite.

Numerical simulations were conducted to design a high-efficiency NH$_3$ decomposition system. The system achieved up to 91% thermal efficiency, validated through ANSYS simulations considering wall surface reactions and turbulent flow. Factors such as NH$_3$ mass flow, operating gauge pressure, and exhaust gas flow were analyzed. The proposed solution aims to address challenges in hydrogen storage and transportation, offering a promising approach for energy applications. Numerical simulations demonstrate the effectiveness of the designed heat recapture NH$_3$ pyrolysis system, achieving up to 94% conversion of NH$_3$ with the use of a nickel catalyst. The rate of NH$_3$ decomposition shows temperature and pressure dependencies, with optimal conditions identified for enhanced reaction rates. Conversion of NH$_3$ remains high within 823 K to 923 K, highlighting the system's efficiency. Comparison between theoretical predictions and simulation results underscores the importance of considering turbulent flow effects in system design.

The proposed solution offers a promising approach to address challenges in hydrogen storage and transportation in the power plant industry. It offers an insight into convective

heat transfer for $NH_3$ pyrolysis. By utilizing $NH_3$ as a hydrogen carrier and integrating its decomposition within gas turbine systems, significant efficiency improvements can be achieved. Further research is needed to optimize system parameters and validate experimental results. Last but not least, the design of a catalyst tube could be one of the approaches to enhance the conversion of $NH_3$ in terms of fluid flow mechanics. One potential approach could be to use swirl tubes to increase the surface area for better convective heat transfer. These findings contribute to advancing clean energy technologies and promoting sustainable power generation.

**Author Contributions:** Conceptualization, methodology, software, analysis, validation, investigation, formal analysis, draft preparation and data curation, J.T.L.; Conceptualization, methodology, investigation, writing—review and editing, supervision and project administration, E.Y.-K.N.; software, writing—review and editing, visualization, and supervision, H.S.; Methodology, supervision and writing—review and editing, H.K.L. All authors have read and agreed to the published version of the manuscript.

**Funding:** This research received no external funding.

**Data Availability Statement:** The datasets presented in this article are not readily available because the data are part of an ongoing study. Requests to access the datasets should be directed to the authors on request.

**Acknowledgments:** The author would like to first express his gratitude to Nanyang Technological University (NTU), Singapore, for the opportunity to study this topic. The author would also like to extend his sincere appreciation to other co-authors for their support.

**Conflicts of Interest:** The authors declare no conflicts of interest.

## Appendix A

The total enthalpy gained by $NH_3$ reported from the numerical simulation is $1.66 \times 10^6$ $Jkg^{-1}$. With the activation energy of 211 $kJmol^{-1}$ or $12,411$ $kJkg^{-1}$, the design could convert 0.134 kg of $NH_3$. The mole ratio of $NH_3 : H_2 : N_2$ obtained from Equation (1) is 2:3:1.

Product mass is calculated by Equation (A1) based on an assumption of full conversion of $NH_3$.

$$\text{Product mass} = \text{mole} \times \text{relative molecular mass} \tag{A1}$$

Product mass fraction is calculated by Equation (A2):

$$\text{Product mass fraction}_i = \frac{\text{mass of product}_i}{\text{total mass of products}} \tag{A2}$$

i : product i.

Percentage difference is calculated to compare between the results obtained in the numerical simulation and the theoretical calculation.

$$\text{Percentage difference} = \left| \frac{\text{RPMF} - \text{CPMF}}{\text{RPMF}} \right| \times 100 \tag{A3}$$

The result obtained from the theoretical calculation is tabulated in Table A1. The percentage differences of $H_2$ mass and $N_2$ mass obtained from the numerical simulation and theoretical calculations are 2.22% and 0.49%, respectively. This could validate the results obtained from the numerical simulation in this study.

**Table A1.** Theoretical Calculation for Mass Fraction of $H_2$ and $NH_3$.

| | Reactant | Product | |
|---|---|---|---|
| | $NH_3$ | $H_2$ | $N_2$ |
| Mole ratio | 2 | 3 | 1 |
| Relative molecular mass ($gmol^{-1}$) | 17 | 2 | 28 |
| Reactant Mass (g) | 134 | - | - |
| Mole (mol) | 7.88 | 11.82 | 3.94 |
| Product Mass (g) | - | 23.64 | 110.32 |
| Calculated Product Mass Fraction (CPMF) | - | 0.176 | 0.824 |
| Reported Product Mass Fraction (RPMF) | - | 0.18 | 0.82 |
| Percentage Difference | - | 2.22% | 0.49% |

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
