# Peer review of "Numerical Simulation of Effective Heat Recapture Ammonia Pyrolysis System for Hydrogen Energy"

_inventions, doi:10.3390/inventions9030056_

Round 1

Reviewer 1 Report

Comments and Suggestions for Authors

TThe submitted manuscript provides very valuable insights into the pyrolysis of ammonia for the utilization of stored hydrogen for gas turbines. Interesting are the solutions regarding the ammonia conversion at different temperature and pressure parameters.

Despite the detailed analysis and visualization of the obtained results, I have several comments of a professional and formal nature:

1) The issue of material embrittlement due to hydrogen is a more complex problem and material embrittlement has not been demonstrated when transporting hydrogen in pipelines at normal temperatures.

2) In the manuscript, different pressure units "bar" and "Pa" are used in the text and figures. It is necessary to use only units from the SI system.

3) On line 36, it is more appropriate to use "%" instead of the word expression "percent".

4) The numbering of mathematical relations is missing.

5) In the "Heat Recapture System" section, they state the heat capacity for hot and cold fluids marked as Ch and Cc. Then they already use the Cmin and Cmax labels, which is misleading and unclear. I propose to improve the description of the use of these labels.

6) It is necessary to check the markings in the part of the text line 179-182. Different designations "k-e", "k-𝜔" and "k-omega" are used here. It is necessary to check these markings and unite them with the marking as stated in "Results and discussion".

7) Please enter the correct name of the ANSYS Chemkin Pro software.

8) On line 289, the authors mention the term "epsilon model". Please enter the correct technical term. Accordingly, it is not clear what the authors meant by this and what it refers to.

9) In the section under Figure 4, the pressures for the reactions are given. Is this the correct notation? It is not clear if it is overpressure related to barometric pressure or if it is absolute pressure. Alternatively, it could also be partial pressure. Similarly, these pressures are shown in Figure 5. It is necessary to improve the text of this part (lines 231-242).

Despite the comments, the manuscript brings valuable knowledge and the obtained results show further possibilities for research.

It is necessary to edit the manuscript according to the comments and improve the text in some parts of the manuscript. It is necessary to unify the notation of pressure units and possibly use temperature in only one temperature scale "°C" or "K". The text of the manuscript uses the designations of ammonia in most cases as the chemical symbol "NH3" but it also often occurs in the word form "ammonia". It is up to the editors to consider whether to keep such different labelling. I have no major objections to it, but perhaps it would be worth considering to unify it.

The results presented in the manuscript are valuable for further research and the content of the manuscript is consistent with the contents of the special issue of the journal.

Reviewer 2 Report

Comments and Suggestions for Authors

The article presents a numerical simulation of an ammonia pyrolysis system with heat recovery.

The design of the work is interesting and innovative. The article is written substantively, reliably, and with particular scientific accuracy. The data presented in the manuscript are interpreted appropriately and coherently, and the conclusions are consistent with the evidence and arguments presented.

However, the manuscript in its current version cannot be accepted and requires several corrections:

1. The abstract should contain the most important achievements of the article. You should summarize the obtained results in a few sentences.

2. About 50% of the literature items are from the last 5 years. The works should be enriched with more current items.

3. Fig. 1 - quality needs to be improved.

4. Fig. 10-12 - Missing units or incorrect marking of units, K and stC used interchangeably, fugures were presented without due care.

5. Fig. 20,24, 26 - similarly as above; the units should be placed next to the axis description. To sum up, most figures need to be corrected so that they reliably present the calculated data.

6. inie 375-378 - bolded unnecessarily

7. Each simulation must be checked under experimental conditions. A computational model that has not been experimentally verified does not guarantee success in experimental, let alone industrial, conditions. If the authors carried out experimental verification, it should be presented at least for one case to prove that the proposed model has a real chance of being used in real conditions and not just assumed ones.

Round 2

Reviewer 1 Report

Comments and Suggestions for Authors

The authors revised the manuscript according to comments and improved some parts of the text. Nevertheless, I have a few comments:

1) On the formal side, it is necessary to check the same notation with a lower index for chemical formulas like "NH3" (such as lines 108, and 122).

2) On line 145, the unit in parenthesis "J/°Cs" is indicated, what does this mean? Shouldn't it be "J/°C" by any chance?

3) Modify the description of the units on the y-axis in Figure 5 as it is used in other figures, for example as in Figure 4.

4) In the text, there are various notations of units such as "kJ/mol" and in another place in the manuscript "kJ.mol-1" and similar other units. It is necessary to unify the notations of the units in the manuscript.

5) Despite the explanation about the used operating pressure, it is necessary to add barometric pressure to the text. From the timeline of actual experimental measurements, it is possible to observe the change in barometric pressure every hour. For this reason, it is better to indicate the absolute (total) pressure instead of the operating pressure.

Comments do not reduce the professional level of the submitted manuscript and are rather of a formal nature. I assume that research in this area will continue and therefore I wish the authors many good ideas and a lot of positive energy.

Author Response

Thank you for your valuable comments. Please see the attachment.

Reviewer 2 Report

Comments and Suggestions for Authors

Thank you for your answers and corrections

Author Response

Thank you for your valuable comments.